# A New Manganese Superoxide Dismutase Mimetic Improves Oxaliplatin-Induced Neuropathy and Global Tolerance in Mice

**DOI:** 10.3390/ijms232112938

**Published:** 2022-10-26

**Authors:** Caroline Prieux-Klotz, Henri Chédotal, Martha Zoumpoulaki, Sandrine Chouzenoux, Charlotte Chêne, Alvaro Lopez-Sanchez, Marine Thomas, Priya Ranjan Sahoo, Clotilde Policar, Frédéric Batteux, Hélène C. Bertrand, Carole Nicco, Romain Coriat

**Affiliations:** 1Institut Cochin, INSERM U 1016 CNRS UMR 8104, Université de Paris, 75005 Paris, France; 2Percy Military Hospital, Gastroenterology, 101 Avenue Henri Barbusse, 92140 Clamart, France; 3Laboratoire des Biomolécules, LBM, Département de Chimie, Ecole Normale Supérieure, PSL University, Sorbonne Université, CNRS, 75005 Paris, France; 4Gastroenterology, Cochin Hospital AP-HP, Université de Paris, 75014 Paris, France

**Keywords:** Pt(IV) prodrugs, superoxide dismutase mimetic, colorectal cancer, oxaliplatin-induced peripheral neuropathy

## Abstract

Reactive oxygen species (ROS) are produced by every aerobic cell during mitochondrial oxidative metabolism as well as in cellular response to xenobiotics, cytokines, and bacterial invasion. Superoxide Dismutases (SOD) are antioxidant proteins that convert superoxide anions (O_2_^•−^) to hydrogen peroxide (H_2_O_2_) and dioxygen. Using the differential in the level of oxidative stress between normal and cancer cells, SOD mimetics can show an antitumoral effect and prevent oxaliplatin-induced peripheral neuropathy. New Pt(IV) conjugate prodrugs (OxPt-x-Mn1C1A (x = 1, 1-OH, 2)), combining oxaliplatin and a Mn SOD mimic (MnSODm Mn1C1A) with a covalent link, were designed. Their stability in buffer and in the presence of sodium ascorbate was studied. In vitro, their antitumoral activity was assessed by the viability and ROS production of tumor cell lines (CT16, HCT 116, KC) and fibroblasts (primary culture and NIH 3T3). In vivo, a murine model of colorectal cancer was created with subcutaneous injection of CT26 cells in Balb/c mice. Tumor size and volume were measured weekly in four groups: vehicle, oxaliplatin, and oxaliplatin associated with MnSODm Mn1C1A and the bis-conjugate OxPt-2-Mn1C1A. Oxaliplatin-induced peripheral neuropathy (OIPN) was assessed using a Von Frey test reflecting chronic hypoalgesia. Tolerance to treatment was assessed with a clinical score including four items: weight loss, weariness, alopecia, and diarrhea. In vitro, Mn1C1A associated with oxaliplatin and Pt(IV) conjugates treatment induced significantly higher production of H_2_O_2_ in all cell lines and showed a significant improvement of the antitumoral efficacy compared to oxaliplatin alone. In vivo, the association of Mn1C1A to oxaliplatin did not decrease its antitumoral activity, while OxPt-2-Mn1C1A had lower antitumoral activity than oxaliplatin alone. Mn1C1A associated with oxaliplatin significantly decreased OIPN and also improved global clinical tolerance of oxaliplatin. A neuroprotective effect was observed, associated with a significantly improved tolerance to oxaliplatin without impairing its antitumoral activity.

## 1. Introduction

Reactive oxygen species (ROS) are highly reactive oxygenated chemical species. All eukaryotic cells, due to their aerobic cellular metabolism, produce ROS. Superoxide anion (O_2_^•−^) is a free radical, toxic to the cell if not regulated, which is transformed into hydrogen peroxide (H_2_O_2_) and dioxygen (O_2_) by an enzyme called superoxide dismutase (SOD). The Manganese superoxide dismutase (MnSOD) is located in the mitochondria and is indispensable for life [1]. Increased oxidative stress occurs due to a disbalance between ROS production and detoxification. Cellular homeostasis is maintained by the presence of detoxification systems such as SODs, catalase, and glutathione (GSH) [2]. ROS production at low intracellular concentrations can lead to cellular proliferation [3,4,5]. When produced at higher levels, they can induce antitumoral activity [6,7]. Indeed, cancer cells were found to have increased sensitivity to oxidative stress compared to normal cells [8] due to a lower detoxification system and higher basal ROS production close to the toxic threshold [9].

Superoxide dismutase mimetics are low-molecular-weight complexes that mimic the activity of superoxide dismutase [10,11,12]. Such compounds may present antioxidant and anti-inflammatory activity [13,14,15,16], and they have also been evaluated for their antitumoral activity [17]. SOD mimetics are proposed to lead to an increase in cellular H_2_O_2_, which induces antitumoral activity in vitro due to the difference in hydrogen peroxide cellular sensitivity between normal and cancer cells [3,6,18,19,20]. In particular, mangafodipir, a contrast agent used in magnetic resonance imaging that concentrates in the liver and displays superoxide dismutase antioxidant properties, has been shown to prevent colorectal tumor growth in mice [21,22]. Oxaliplatin is a major drug in digestive oncology, in adjuvant [23] or metastatic situations [24,25,26,27], in colorectal cancer, and in oesogastric cancer, pancreatic adenocarcinoma, and bile duct carcinoma. Platinum derivatives are known to form adducts with the DNA of cells, thus inducing cell cycle blockade and cell apoptosis. This chemotherapy leads to an oxidative burst in the cancer cell, which is part of its antitumoral activity [6] but is also associated with side effects [28]. It has been shown that mangafodipir can prevent and/or relieve oxaliplatin-induced neuropathy in cancer patients [21].

Oxaliplatin-induced peripheral neuropathy occurs in >90% of treatments, is responsible for chronic pain and disability, and is dose-limiting [29]. It has been shown that oxaliplatin-induced peripheral neuropathy is induced by the oxidative burst and direct toxicity of O_2_
^−^ in the dorsal ganglion [28]. 

Mangafodipir, as a MnSOD mimetic, has been shown as neuroprotective when co-administrated with oxaliplatin, leading to a significant decrease of oxaliplatin-induced peripheral neuropathy in mice in a phase I trial [21]. In Coriat et al., 22 cancer patients with grade ≥ 2 oxaliplatin-associated peripheral neuropathy received intravenous mangafodipir following oxaliplatin [21]. In 77% of cases, oxaliplatin-induced neurotoxicity improved or stabilized after four cycles. After eight cycles, oxaliplatin-induced neurotoxicity was downgraded in six out of seven patients. Glimelius et al. [30] conducted a phase II study with 179 colorectal cancer patients with oxaliplatin-induced peripheral neuropathy, randomized between calmangafodipir and placebo. Calmangafodipir-treated patients had significantly less cold allodynia (mean 1.6 versus 2.3, *p* < 0.05) and significantly fewer sensory symptoms in the Leonard scale (cycle 1–8 mean 1.9 versus 3.0, *p* < 0.05 and during follow-up after 3 and 6 months, mean 3.5 versus 7.3, *p* < 0.01). Response rate, progression-free, and overall survival did not differ among groups. A phase III study (POLAR study) comparing PledOx (calmangafodipir) to placebo in patients with oxaliplatin-induced peripheral neuropathy was shut down early because PledOx 5 μmol/kg in combination with chemotherapy did not reduce the risk of moderate to severe chemo-induced peripheral neuropathy (CIPN) at 9 months after the first cycle. Mangafodipir commercialization stopped, and the manganese atom was replaced by gadolinium as a contrast MRI agent. Thus, mangafodipir availability has been impaired. Moreover, the overall negative charge of mangafodipir, impairing its cellular penetration, has been suspected for the lack of antitumoral efficacy in vivo [9].

Mn1 [31] is a MnSOD mimic bio inspired by the active site of MnSOD, specific of superoxide, [32] with better intracellular diffusion compared to mangafodipir [13,15,16]. It was shown to exert anti-inflammatory activity in a cellular model of inflammatory bowel diseases (HT29-MDM2) associated with an antioxidant effect [13]. Guillaumot et al. studied this complex (called MAG in the study) in vitro and in vivo. In vitro, MAG induced hydrogen peroxide production in tumoral cells (HT 29, CT 26) and embryonic fibroblasts (NIH 3T3). A cytotoxic additive effect in combination with oxaliplatin (oxaliplatin + Mn1 treatment) was observed on cancer cells. In vivo, in a murine model of colorectal cancer, Mn1 (MAG) in association with oxaliplatin did not have a better antitumoral effect than oxaliplatin but showed a significant decrease in peripheral neuropathy induced by oxaliplatin [33].

In this work, we developed and evaluated a panel of new chimeric molecules in the form of Pt(IV) prodrugs that covalently link oxaliplatin with a MnSOD mimetic derived from Mn1—Mn1C1A. Pt(IV) prodrugs constitute one of the alternative strategies to combination treatments devised to achieve, using appropriate axial moieties, targeted delivery, reduced side effects and toxicity, increased anticancer activity, or improved bioavailability [34,35]. Pt(IV) complexes are reduced in cells or tumor microenvironments, releasing the active parent Pt(II) anticancer agent and axial moieties. A covalent conjugation between the two active components is expected to present several advantages compared to combination treatment, such as the possible control of the ratio of active components penetrating the cells. Three conjugates were synthesized and characterized (Figure 1). We assessed the cytotoxic activity of these new compounds in vitro and, for one of them, the protective effect against oxaliplatin-induced peripheral neuropathy and antitumoral activity in vivo on a murine model of colorectal cancer.

## 2. Results

### 2.1. Synthesis of 1C1A, Pt(IV) Conjugates and the Corresponding Mn Complexes

The 1C1A ligand of the MnSOD mimic Mn1C1A was prepared starting from ligand 1 of the SOD mimic Mn1 (MAG), already described in ref. [31]. Ligand 1 was alkylated by reductive amination with ethylglyoxalate in the presence of sodium cyanoborohydride to afford the desired ester after HPLC purification. The 1C1A ligand was obtained after ester hydrolysis with excess sodium hydroxide and further HPLC purification (Figure 1).

For synthetic purposes, oxaliplatin (Pt^II^(dach)(ox)) was prepared by a classical procedure in three steps (see Appendix A) with a 74% overall yield starting from potassium tetrachloroplatinate: chloride ligands were exchanged for iodide, coordination was established with the (1R,2R)-cyclohexane-1,2-diamine (dach) ligand, and the complex formed was treated with freshly prepared silver oxalate [36,37]. Oxidation of oxaliplatin in the presence of aqueous hydrogen peroxide alone or in acetic acid afforded the Pt(IV) intermediates Pt^IV^(dach)(ox)(OH)_2_ (73% yield) and Pt^IV^(dach)(ox)(OAc)(OH) (71% yield) [38]. The Pt(IV) conjugates were obtained by ester coupling between 1C1A and a Pt(IV) intermediate in different conditions (Figure 1). The bis-conjugate compound with the MnSOD mimic ligand in both axial positions, OxPt-2-1C1A, and the mono-OH conjugate OxPt-1OH-1C1A, were prepared by reaction of Pt^IV^(dach)(ox)(OH)_2_ with 2.1 and 1.2 equivalents of 1C1A using TBTU (2-(1H-benzotriazole-1-yl)-1,1,3,3-tetramethylaminium tetrafluoroborate) as a coupling agent in the presence of Et_3_N in DMF. The mono-OAc conjugate OxPt-1-1C1A was obtained using Pt^IV^(dach)(ox)(OAc)(OH) and 1C1A with DCC (N,N′-dicyclohexylcarbodiimide) as coupling agent. The three conjugates were purified by preparative reverse-phase HPLC and fully characterized (Appendix A).

Mn(II) coordination was performed in situ before each experiment using a solution of MnCl_2_ and the titrated conjugate-ligand in desired ratios. The three compounds OxPt-x-1C1A (x = 1OH, 1, 2) were titrated by UV-visible spectrophotometry as already described [13,15] using a solution of ZnCl_2_, known to bind to ligand 1 with higher affinity than Mn^2+^ and being redox inactive (Appendix A).

The hydrolytic stability of the conjugate ligands OxPt-x-1C1A (x = 1OH, 1, 2) and of the corresponding complexes OxPt-x-Mn1C1A (x = 1OH, 1, 2) obtained via in situ Mn coordination was evaluated in buffer (HEPES buffer 50 mM at pH = 7.4) by analytical HPLC. The compounds OxPt-x-1C1A (x = 1OH, 1, 2) were stable in buffer for more than 24 h, with no sign of hydrolysis [39,40]. Instability of the compounds in the presence of Mn(II) salt (1 equivalent for x = 1OH,1, and 2 equivalents for x = 2) was observed with half-lives of the complexes (time to lose 50% of the signal of the compound) of ≈30 min for OxPt-1OH-Mn1C1A and OxPt-1-Mn1C1A and 4 h for OxPt-2-Mn1C1A (Appendix A). The stability of the compounds was then evaluated in the presence of a cellular reductant (excess of sodium ascorbate) to reflect the kinetics of reduction of the Pt(IV) conjugates to Pt(II) oxaliplatin in cells (Appendix A). Half-lives of reduction for the Pt(IV) compounds OxPt-1OH-1C1A, OxPt-1-1C1A, and OxPt-2-1C1A were found to be ≈7 h, >24 h and ≈7 h and in line with data from the literature on Pt(IV) conjugate hydroxydo/acetato- type ligands [39,40,41,42]. The same experiment performed on the corresponding complexes OxPt-x-Mn1C1A (x = 1OH, 1, 2) directly after in situ Mn coordination indicated a rapid disappearance of the compounds’ HPLC signature with half-lives of ≈1 h, 30 min, and ≈2.5 h, respectively (Appendix A).

Further studies by UV-Vis spectroscopy (titration with MnCl_2_) and mass spectrometry (see Appendix A for full description, Appendix A) lead us to rationalize the reactivity of the Pt(IV) conjugates upon Mn coordination as follows: upon coordination of the ligand with Mn^2+^, the Pt-O bond of the acetate-type ligand is cleaved through hydrolysis with coordination of the carboxylato group to the metal ion (Mn), forming a favorable five-membered metallacycle and generating a six-coordinated oxidized Mn^III^ complex. Although no signature from a Pt species could be unambiguously assigned on the HPLC chromatograms or by MS spectrometry (see Appendix A), the hydrolysis prompted by Mn coordination is expected to produce the Pt(IV) intermediates Pt(dach)(ox)(OH)_2_ in the case of OxPt-1OH-Mn1C1A and OxPt-2-1C1A conjugates and Pt(dach)(ox)(OAc)(OH) for the OxPt-2-Mn1C1A conjugate. Further hydrolysis of the equatorial oxalate ligand may occur [39]. Detailed mechanistic investigations were not pursued (*vide infra*).

The intrinsic SOD activity, or activity outside of any cellular context, refs. [12,13,32] of the new MnSOD mimic Mn1C1A (prepared in situ by Mn coordination of the titrated ligand 1C1A, see Appendix A) and of the freshly prepared OxPt-x-Mn1C1A (x = 1OH,1,2) conjugates was determined by the McCord–Fridovich assay, as previously described in refs. [12,32]. The parent MnSOD mimic Mn1 was included as a reference [13,43]. The results provided as k_McCF_ are presented in Table 1. The SOD activity of the new complex Mn1C1A was found to be higher than that of the parent complex Mn1, with k_McCF_ of 3.11 × 10^8^ M^−1^ s^−1^, indicating a favorable impact of the additional carboxylate group.

Although this first generation of Pt(IV) oxaliplatin–MnSOD mimic conjugates have limited stability upon Mn coordination, they generate mixtures composed of the Pt(IV) conjugate, the MnSOD mimic Mn1C1A, and a Pt(IV) species (Pt(dach)(ox)(OH)_2_ or Pt(dach)(ox)(OAc)(OH)) that generates oxaliplatin upon reduction [41]. They may present interesting biological activity, keeping in mind that, in the time scale of the cellular experiments (incubation of 24–72 h in the extracellular medium), the conjugates will be completely hydrolyzed in the medium [39,42]. The three Pt(IV) conjugates were further evaluated in cells along with the association condition Mn1C1A + oxaliplatin and oxaliplatin alone. 

### 2.2. ROS Levels In Vitro

The Pt(IV) conjugates and Mn1C1A associated with oxaliplatin increased H_2_O_2_ levels in all cell lines. Figure 2 shows the mean H_2_O_2_ levels measured in HCT 116, CT26, NIH 3T3, and KC cells after treatment with the three Pt(IV) conjugates, oxaliplatin alone, Mn1C1A + oxaliplatin, or vehicle (at 2.5 μM, 5 μM, and 10 μM) for 24 h. In HCT 116 cells, treatment with Mn1C1A + oxaliplatin significantly increased the mean H_2_O_2_ levels (*p* < 0.01) compared to oxaliplatin alone with all tested concentrations. In CT26 cells, treatment with OxPt-2-Mn1C1A and Mn1C1A + oxaliplatin significantly increased the mean H_2_O_2_ levels (*p* < 0.05) compared to oxaliplatin alone with all tested concentrations. In NIH 3T3 cells, treatment with OxPt-1OH-Mn1C1A, OxPt-1-Mn1C1A, OxPt-2-Mn1C1A, and Mn1C1A + oxaliplatin significantly increased the mean H_2_O_2_ levels (*p* < 0.001) compared to oxaliplatin alone with all tested concentrations. In KC cells, treatment with Mn1C1A + oxaliplatin significantly increased the mean H_2_O_2_ levels (*p* < 0.01) compared to oxaliplatin alone with all tested concentrations.

Mean O_2_^.−^ levels in all cell lines were measured after 24 h of treatment exposure. O_2_^.−^ levels were significantly higher in HCT 116 and NIH 3T3 cell lines treated with Mn1C1A + oxaliplatin than oxaliplatin alone. In HCT 116 cells, oxaliplatin + Mn1C1A significantly increased O_2_^.−^ levels compared to oxaliplatin alone (*p* < 0.05). In CT26 and KC cells, no significative difference was observed in O_2_^.−^ levels. In NIH 3T3 cells, OxPt-2-Mn1C1A and oxaliplatin + Mn1C1A significantly increased O_2_^.−^ levels compared to oxaliplatin alone (*p* < 0.05) (Appendix A). GSH cellular levels, reflecting the detoxification system, were stable in CT26, HCT 116, and NIH 3T3 and lowered in KC cells treated with MnSOD mimetics (Appendix A).

### 2.3. In Vitro Cell Viability

Cell viability was assessed by crystal violet on day 2 after 24 h of treatment (3 Pt(IV) compounds, oxaliplatin alone, and Mn1C1A + oxaliplatin or vehicle, at 2.5 μM, 5 μM, and 10 μM) (Figure 3). In vitro antitumoral activity was significantly higher in tumoral cell lines treated with Pt(IV) conjugate compounds and Mn1C1A associated with oxaliplatin compared to oxaliplatin alone. The antitumoral effect was proportional to H_2_O_2_ levels.

Cell viability assessed by crystal violet was also measured on a primary culture of murine fibroblasts (Figure 4). The Pt(IV) conjugates and oxaliplatin + Mn1C1A had significantly higher cytotoxicity on primary fibroblast culture cells than oxaliplatin alone. The MnSODm Mn1C1A alone was not cytotoxic on fibroblasts.

### 2.4. In Vivo Antitumoral Effect

In vivo, a murine model of colorectal cancer was created with subcutaneous injection of CT26 cells. After tumor growth, Balb/c mice underwent randomization in four groups: vehicle, oxaliplatin, the bis-conjugate OxPt-2-Mn1C1A, and oxaliplatin associated with Mn1C1A. The tumoral volume was followed during treatment (Figure 5).

In the control group treated with the vehicle, tumoral growth was exponential. Mice treated with OxPt-2-Mn1C1A had significantly slower tumoral growth than the control group but significantly faster tumoral growth than mice treated with oxaliplatin alone (*p* < 0.05). The association of Mn1C1A to oxaliplatin did not affect oxaliplatin’s antitumoral activity. Mice treated with Mn1C1A + oxaliplatin had the same and highest tumoral response compared to OxPt-2-Mn1C1A and control.

### 2.5. Treatment Tolerance In Vivo

The clinical tolerance of the treatments was assessed with a clinical score composed of four items (weight loss (0–2), alopecia (0–2), diarrhea (0–1), and asthenia (0–2)) measured in all mice treated with the vehicle, oxaliplatin 10 mg/kg weekly (IP), Mn1C1A + oxaliplatin 10 mg/kg weekly (IP), and OxPt-2-Mn1C1A 10 mg/kg weekly (IP) (Figure 6) The higher the score, the poorer the clinical tolerance. 

Mice treated with oxaliplatin alone had significantly poorer clinical tolerance than mice treated with Mn1C1A + oxaliplatin. Thus, 80% of mice treated with oxaliplatin could not receive their second injection due to severe weight loss, but every injection could be received in the Mn1C1A + oxaliplatin group with no significant weight loss.

Mice treated with oxaliplatin alone had a high mean clinical score [4] after one intraperitoneal injection, and 80% could not receive the second injection due to weight loss of >20%. In the vehicle, OxPt-2-Mn1C1A, and Mn1C1A + oxaliplatin groups, each mouse received weekly injections. Figure 6A shows the mean clinical score during the whole experiment. Mice treated with Mn1C1A + oxaliplatin had a significantly lower clinical score showing better clinical tolerance of the treatment (*p* < 0.01) than mice receiving oxaliplatin alone with the same dose. Mice treated with the conjugate OxPt-2-Mn1C1A also had a significantly lower clinical score than mice treated with oxaliplatin alone (*p* < 0.05). Figure 6B shows the clinical score evolution over time.

Mice treated with oxaliplatin + Mn1C1A showed significant less hypoalgesia induced by oxaliplatin than mice treated with oxaliplatin alone. The Von Frey test reflecting chronic hypoalgesia, with force (grams) necessary to induce paw withdrawal, was performed before treatment and during the whole experiment (Figure 7). Figure 7A shows the mean Von Frey score during the whole experiment in mice treated with the vehicle, oxaliplatin, Mn1C1A + oxaliplatin, and OxPt-2-Mn1C1A. In the Mn1C1A + oxaliplatin group, significantly less hypoalgesia occurred than in the oxaliplatin alone group (mean: 2.57 g vs. 3.95 g, *p* < 0.05). The difference between OxPt-2-Mn1C1A and oxaliplatin alone was not significant. Figure 7B shows the Von Frey score evolution over time (days). Mice treated with oxaliplatin had progressive chronic hypoalgesia with an increase in the force necessary to withdraw their paw. Mice treated with the same dose of oxaliplatin associated with Mn1C1A exhibited the same behavior as mice treated with the vehicle.

### 2.6. ROS Production In Vivo

The mean advanced oxidation protein products (AOPP) level in mice serum was measured after sacrifice (Figure 8). Oxidative stress represented by seric AOPP was significantly lower in the Mn1C1A + oxaliplatin and OxPt-2-Mn1C1A groups than in the oxaliplatin group (*p* < 0.001 and <0.05, respectively), reflecting the MnSOD activity of the molecules.

### 2.7. VEGF Production In Vivo

After sacrifice, the tumoral RNA of VEGF was measured with PCR (Figure 9). Figure 9A represents the PCR VEGF/βactin in tumoral tissue of mice treated with either the vehicle, oxaliplatin, OxPt-2-Mn1C1A, or Mn1C1A + oxaliplatin, showing a significantly higher production of VEGF RNA in the Mn1C1A + oxaliplatin group.

### 2.8. Autophagy Markers In Vivo

Figure 9B represents the autophagy marker LC3 measured in Western blot, showing a significantly higher production of the LC3 protein in the Mn1C1A + oxaliplatin group. There was no difference in SQSTM1 production among the groups. 

### 2.9. Histological Analysis

No significant difference in histological tumoral analysis was found between the groups. Caspase 3 expression in immunohistochemistry was dependent on tumoral response. Luxol fast blue coloration in paw skin showed no significant difference between the groups. 

## 3. Discussion

We designed and synthesized the first Pt(IV) complexes as covalent conjugates between oxaliplatin and MnSOD mimics derived from complex Mn1 (MAG) [13,31,43] to evaluate them for their antitumoral activity and effect on oxaliplatin-induced peripheral neuropathy. Three Mn-uncoordinated Pt(IV) conjugate OxPt-x-1C1A (x = 1OH,1,2) were prepared and characterized. In situ, with half-lives t_1/2_ between 30 min and 2 h, Mn coordination induces hydrolysis and cleavage of the Pt-O linkage bond with the folding of the carboxylate arm onto the Mn ion, releasing the Mn1C1A MnSODm and a Pt species derived from oxaliplatin (Appendix A). This reactivity is proposed to arise from the favourable formation of a five-membered metallacycle between the carboxylate group and the Mn ion [44]. Conjugates with suitable hydrolysis stability and stability upon Mn coordination can be prepared by increasing the carbon linker length, which supports our hypothesis. Results on this second generation of conjugates will be reported in due course. Therefore, the in situ preparation of the OxPt-x-Mn1C1A (x = 1OH,1,2) complexes for the cellular assays and in vivo injections lead to a mixture of compounds that depend on the time scale of the experiment: Pt(IV) conjugate, Mn1C1A, and a Pt(IV) species eventually leading to an oxaliplatin derivative. This first series of Pt(IV) conjugates were further evaluated in vitro and in vivo in the case of the bis-conjugate.

In vitro, the three Pt(IV) conjugate OxPt-x-Mn1C1A (x = 1OH,1,2) and the Mn1C1A MnSODm associated with oxaliplatin showed a significative rise of H_2_O_2_ cellular levels associated with superoxide dismutase activity in tumoral and fibroblasts cell lines compared to oxaliplatin alone. This increase could be associated with the SOD activity of Mn1C1A. This effect was proportional to the in vitro antitumoral activity with the strongest cytotoxicity for the association of Mn1C1A + oxaliplatin. This result confirms our hypothesis of an oxidative burst maximizing the antitumoral activity of oxaliplatin and is the same effect as observed by Guillaumot et al. [33] with Mn1 (MAG) and Coriat et al. [21] with mangafodipir. The O_2_^.−^ levels were not lower in cells treated with MnSODm, contrary to what was expected. This result may be explained by the delay of the measure (24 h after treatment), which is potentially late for an O_2_^.−^ change in cellular content [45].

Although the Mn1C1A alone was not cytotoxic on primary fibroblasts, the conjugate compounds showed higher toxicity in fibroblasts and NIH3T3 cells than oxaliplatin alone. This result is surprising due to the fact that tumoral cells show a lower cytotoxic threshold of H_2_O_2_ than normal cells. This result can be explained by the high rise of H_2_O_2_ levels_,_ with OxPt-x-Mn1C1A (x = 1OH,1,2) exceeding all thresholds for innocuity. This result contrasts with the good in vivo tolerance of the conjugate compounds (clinical, haematological). 

In vivo, in the model of murine colorectal cancer with subcutaneous injection of CT26 cells, the association of Mn1C1A to oxaliplatin did reduce the antitumoral activity of oxaliplatin but did not maximize its antitumoral activity. The same phenomenon was observed by Guillaumot et al. [33] with Mn1 (MAG). The intra-tumoral penetration of the compound is not certain, but the low levels of AOPP in the mice sera reflect seric MnSOD activity in vivo. This parallels the reduction in the protein MnSOD expression in inflamed cells incubated with Mn1 in comparison with non-treated inflamed cells [13]. For the OxPt-2-Mn1C1A bis-conjugate compound, lower antitumoral activity than oxaliplatin alone was observed with no significant neuroprotective effect. The lower anti tumoral activity of OxPt-2-Mn1C1A could be explained by the lower relative dose of Platinum received: in oxaliplatin, 10 mg·kg^−1^ = 4.91 mg·kg^−1^ of Pt, in oxaliplatin and Mn1C1A, 10 mg·kg^−1^ of each compound = 4.91 mg·kg^−1^ of Pt and 1.095 mg·kg^−1^ of Mn but in OxPt-2-Mn1C1A, 10 mg·kg^−1^ = 1.39 mg·kg^−1^ of Pt and 0.785 mg·kg^−1^ of Mn. Furthermore, intratumoral and intracellular concentrations of Pt and/or Mn in the case of OxPt-2-Mn1C1A and oxaliplatin + Mn1C1A treatments should be quantified to compare the penetration and the capacity to reach the tumor site.

A possible explanation for the lack of additive antitumoral effect of the combination treatment in vivo is the vascular protective effect of MnSOD mimetics, with an increase in VEGF production, which has already been shown with these molecules [46,47]. We established a significant rise in VEGF production with the MnSODm Mn1C1A associated with the oxaliplatin group. VEGF production by tumoral cells participates in tumor growth and proliferation [48]. With increasing VEGF production and thus neoangiogenesis, stimulation could impair the additional antitumoral activity of the MnSOD mimic. An association between the Mn1C1A or the conjugate OxPt-2-Mn1C1A and an anti-VEGF as already used in colorectal cancer, such as bevacizumab or aflibercept, could be interesting in further studies. 

The role of autophagy in tumoral response is crucial in colorectal cancer. It has been shown in human colorectal tumors that overexpression of Beclin-1 determines a poor prognosis factor with lower efficacy of chemotherapy [49]. Oxaliplatin inhibits autophagy in colorectal cancer with microsatellite stability [50]. In contrast, the oxidative burst induces autophagy, as shown in the literature [51,52] and the experiment with a higher LC3 tumoral production in the Mn1C1A + oxaliplatin group [53]. An association of the MnSODm with an autophagy inhibitor could be interesting to maximize the antitumoral activity of the MnSODm. 

In vivo, the association of Mn1C1A with oxaliplatin significantly reduced oxaliplatin-induced peripheral neuropathy compared to the oxaliplatin group. This neuroprotective effect was shown in mice and humans with MnSOD mimetics [21,33] with protection from demyelination. The significant efficacy of the Mn1C1A MnSODm in mice should therefore be confirmed in phase I trials associated with oxaliplatin. 

The association of Mn1C1A and oxaliplatin is the first to have shown a better global clinical tolerance of oxaliplatin with a clinical score based on weight loss, fatigue, diarrhea, and alopecia. Mice receiving Mn1C1A with the same full dose of oxaliplatin as mice receiving oxaliplatin alone had significantly better clinical tolerance of the treatment and therefore received more oxaliplatin at the end of the experiment compared to mice receiving oxaliplatin alone of whom 80% did not receive the second injection due to significant weight loss. 

The improvement of clinical tolerance without impairing antitumoral efficacy associated with a neuroprotective effect raises the oxaliplatin cumulative dose tolerated and, therefore, could improve its antitumoral efficacy. In the study by Guillaumot et al. [33], Mn1 (MAG) showed significant in vitro tumoral cytotoxicity, a neuroprotective effect in vivo but no antitumoral activity in vivo. Clinical tolerance was not assessed. 

In conclusion, this new MnSOD mimetic Mn1C1A, when associated with oxaliplatin, improves clinical tolerance of oxaliplatin in vivo and prevents oxaliplatin-induced peripheral neuropathy without impairing its antitumoral activity. Given these results, this new MnSOD mimetic molecule shows promising perspectives with clinical relevance and should be confirmed in a human phase I clinical trial. This Pt(IV) prodrug strategy combining oxaliplatin and MnSODm in a covalent conjugate will be further evaluated during a second-generation study with optimized properties. 

## 4. Material and Methods

### 4.1. Ligand and Conjugate Compounds

A detailed description of the synthesis and characterizations of the intermediates, 1C1A and MnSOD complex Mn1C1A and OxPt-x-1C1A (x = 1OH,1,2) conjugates, UV-Vis titrations, HPLC experiments, MS spectrometry experiments with direct infusion, and McCord–Fridovich assays is available in the Appendix A.

The chemical formula and molecular weights of the compounds studied in this work are as follows: Mn1C1A C_21_H_26_MnN_6_O_3_ with a molecular weight of 465.14 g/mol, OxPt-1OH-Mn1C1A (C_29_H_41_ClMnN_8_O_8_Pt) with a molecular weight of 914.18 g/mol (associated ligand OxPt-1OH-1C1A: C_29_H_42_N_8_O_8_Pt, 825.78 g/mol), OxPt-1-Mn1C1A (C_31_H_43_ClMnN_8_O_9_Pt) with a molecular weight of 957.20 g/mol (associated ligand OxPt-1-1C1A: C_31_H_44_N_8_O_9_Pt, molecular weight 867.81 g/mol), and OxPt-2-Mn1C1A (C_50_H_66_Cl_2_Mn_2_N_14_O_10_Pt), with a molecular weight of 1397.29 g/mol (associated ligand OxPt-2-1C1A: C_50_H_68_N_14_O_10_Pt, molecular weight 1220.25 g/mol). The three OxPt-x-Mn1C1A (x = 1OH,1,2) complexes and Mn1C1A were prepared in situ by mixing the corresponding titrated ligands (OxPt-x-1C1A (x = 1OH,1,2) and 1C1A) in aqueous solutions with a solution of MnCl_2_ (M = 125.84 g·mol^−1^) in HEPES (100 mM, pH 7) at a 4 mM concentration at the desired ratio: 1:1 molar ratio (ligand:Mn) for Mn1C1A, OxPt-1OH-1C1A, and OxPt-1-1C1A and 1:2 molar ratio for OxPt-2-1C1A.

Oxaliplatin used in the in vitro and in vivo experiments was obtained from Accord healthcare^®^ (Lille, France, flask of 200 mg/40 mL).

### 4.2. Cell Cultures and Treatments

All immortalized cell lines were routinely tested for mycoplasma. All cell lines were purchased from the ATCC. Mouse embryonic fibroblast cells (NIH 3T3), mouse colorectal cancer cells (CT26), mouse pancreatic cancer cells (KC), and human colorectal cancer cells (HCT 116) were grown in Dulbecco’s Modified Eagle Medium high glucose 500mL (Thermofisher^®^, Waltham, MA, USA) with 10% Fetal bovine serum (Thermofisher^®^) and 5mL penicillin/streptomycin (Thermofisher^®^). The primary fibroblast cell culture was isolated from mouse skin and grown in the same culture medium.

### 4.3. ROS Levels In Vitro

When grown to confluence, the different cell lines were treated with trypsin (Thermofisher^®^), counted, and seeded in flat bottom 96-well plates (Corning^®^, New York, NY, USA). CT 26, KC, and HCT 116 cells were seeded with 10^5^ cells/well, and NIH 3T3 with 5. 10^4^ cells/well. A decreasing range was made for each cell line. After 24 h, experimental treatment was applied in triplicate with different concentrations: 2.5 μM, 5 μM, and 10 μM of oxaliplatin, Mn1C1A + oxaliplatin, OxPt-1OH-Mn1C1A, OxPt-1-Mn1C1A, OxPt-2-Mn1C1A, or control (culture medium). After 24 h of treatment exposure, cellular levels of H_2_O_2_, GSH, and O_2_^.−^ were assessed by spectrofluorimetry with 2–7 di-chlorodihydroFluorescein diacetate (H_2_-DCFDA), monochlorobimane, and dihydroethidium (DHE) (D23107, Thermofisher^®^, Waltham, MA, USA), respectively. Cells were washed with Phosphate Buffer saline (PBS) and incubated in a dark environment with 100 μL per well of 5 μM H_2_-DCFDA (D399, Thermofisher^®^, Waltham, MA, USA), 15 μM DHE, or 5 μM monochlorobimane (M1381MP, Thermofisher^®^, Waltham, MA, USA) diluted in PBS. ROS levels were assayed by spectrofluorimetry on a Fusion spectrofluorimeter (PerkinElmer, Waltham, MA, USA.), and fluorescence intensity was recorded every hour over a period of 6 h. Crystal violet viability was assessed at the same time for each cell line, and ROS production was calculated with subtraction of fluorescence between hour 6 and hour H0 reported to crystal violet viability. Student t- and ANOVA tests were performed. Data are presented as the mean results of all concentrations tested.

### 4.4. Cell Viability In Vitro

When grown to confluence, the different cell lines were treated with trypsin (Thermofisher^®^), counted, and seeded in flat bottom 96-well plates (Corning^®^ New York, NY, USA). CT 26, KC, and HCT 116 cells were seeded with 10^5^ cells/well, and NIH 3T3 with 5 × 10^4^ cells/well. A decreasing range was determined for each cell line. After 24 h, experimental treatment was applied in triplicate with different concentrations: 2.5 μM, 5 μM, and 10 μM of oxaliplatin, Mn1C1A + oxaliplatin, OxPt-1OH-Mn1C1A, OxPt-1-Mn1C1A, OxPt-2-Mn1C1A, or control (culture medium). For primary fibroblasts, 10^4^ cells per well were seeded, and the experimental treatment was applied after 48 h, with 2.5 μM, 5 μM, and 10 μM of oxaliplatin, Mn1C1A + oxaliplatin, OxPt-1OH-Mn1C1A, OxPt-1-Mn1C1A, OxPt-2-Mn1C1A and Mn1C1A alone, or control. For all cells, the treatment was removed from the well after 24 h and replaced by a cell medium. Cell viability was assessed 24 h later by crystal violet assay and read with UV spectrometry (Fusion, PerkinElmer). Data are presented as the mean results of all concentrations tested.

### 4.5. Animals

The mice model was performed on Balb/c female mice, aged 6 weeks, purchased from Janvier Labs^®^ (Rte du Genest, 53940 Le Genest-Saint-Isle, France). All animals were housed in ventilated cages, from 7 to 10 per cage, with rodent diet and water ad libitum (Teklad global 16% protein, Envigo^®^ Indianapolis, IN, USA). All animals were exposed to a standard light cycle of 12 h on and 12 h off. All efforts were pursued to minimize animal distress and to reduce the number of animals used. Mice were euthanized by cervical dislocation under isoflurane+0_2_ anesthesia. All animal manipulations were presented and approved by the Ethics Committee of Paris University (Saisine #8394). All experiments were performed in accordance with European and French institutional guidelines. 

### 4.6. Colorectal Cancer Mice Model

A subcutaneous injection of 1.5 × 10^6^ CT26 cells was performed on female Balb/c mice aged 6 weeks. Until a measurable tumoral growth, mice were randomized into 4 groups with similar mean tumor sizes. One group of 8 mice received oxaliplatin 10 mg/kg/week by intraperitoneal injection, a group of 9 mice received OxPt-2-Mn1C1A 10 mg/kg/week by intraperitoneal injection, 9 mice received Mn1C1A + oxaliplatin 10 mg/kg/week by intraperitoneal injection, and 8 mice received vehicle once a week by intraperitoneal injection. The injection was cancelled if weight loss was superior to 20% of the initial weight.

Tumor growth was assessed with a numeric caliper once a week for 4 weeks. Sacrifice occurred after a threshold of maximum tumor size in order to comply with ethical guidelines. Tumor volume was calculated as follows: TV (mm^3^) = (L × W^2^)/2, where L is the longest and W is the shortest radius of the tumor in millimeters. Results were expressed as means ± standard deviation of tumor volumes. Data were analyzed with the Cox and Anova models. 

### 4.7. Clinical Score

A clinical score based on 4 items: diarrhea (scoring 0 = absence to 1 = presence), fatigue (0 = normal activity, 1 = diminution of locomotor activity, 2 = prostration), alopecia (0 = none, 1 = mild, 2 = severe), and weight loss (0 = none, 1 = 10–20%, 2 ≥ 20%) was calculated by 2 different investigators, twice a week. 

### 4.8. Oxaliplatin-Induced Peripheral Neuropathy Assessment

Chronic oxaliplatin-induced peripheral neuropathy was assessed with a Von Frey Test [33,54,55]. The Von Frey test assesses chronic hypoesthesia induced by oxaliplatin peripheral neuropathy. Mice were accustomed to the test twice a week for 2 weeks before tumoral injection. The Von Frey test is standardized with the same experimental conditions to limit stress on animals. Mice are placed on a grid. After 5 min, when the mouse is calm and motionless, a hind paw is touched with the tip of a flexible fiber of a given length and diameter. The fiber is pressed on the plantar surface with a vertical force. Once the fiber bends, the force applied is maximal. Thus, a reproducible force can be applied by the investigator. It is known that rodents withdraw their paw as soon as it is touched [56]. The Von Frey standardized kit includes filaments with bending force from 0.008 g to 300 g. Each filament was tested from the lower force until the mouse pulled back its paw, considering the Von Frey test positive. Paw movements associated with locomotion were not counted as a withdrawal response. The control group (receiving vehicle) was considered a reference for normal Von Frey values.

### 4.9. Samples Analysis

After sacrifice, mice blood was analyzed to assess blood cell count, renal function (urea), and alanine aminotransferase. Advanced oxidation protein products (AOPP) on blood samples were measured by spectrophotometry. The assay was calibrated using chloramine-T. The absorbance was read at 340 nm on a microplate reader (Fusion, PerkinElmer). AOPP concentrations were expressed as μmol/L of chloramine-T equivalents [57].

Polymerase chain reaction (PCR) for VEGF (Vascular Endothelial Growth Factor) was processed with tumoral samples of each group. Total mRNA was extracted from crushed tumoral samples with Trizol reagent (Invitrogen^®^, Waltham, MA, USA). The qRT-PCR was performed with a Quanti Tect SYBR^®^ Green RT-PCR kit on a Light Cycler 480 II instrument (Roche Applier Science, Meylan, France). We used RTQ RT–PCR for the relative quantification of VEGF mRNA in tumor specimens, using β-actin mRNA as an internal control. The relative fold-change was calculated for the reference group using the formula −2^ΔΔCt^.

Autophagy assessment with measurement of LC3 and SQSTM1 production was performed through Western blots on the tumoral samples of each group. Tumor cells were lysed in ice-cold RIPA buffer, supplemented with 25 mmol/L sodium fluoride, anti-protease 1%, and 0.5 mmol/L sodium orthovanadate. Equal amounts of proteins were loaded and separated by 10% sodium dodecyl-sulfate-polyacrylamide gel electrophoresis (45101023Biorad^®^, Gémenos, France). After transfer and blocking with 5% fat-free milk and 0.1%tween in PBS, the nitrocellulose membrane was incubated overnight at 4 °C with a 1:1000 dilution of an anti-mouse SQSTM1 monoclonal antibody (20839, Invitrogen, Thermofisher^®^, Waltham, MA, USA) and a 1:1000 dilution of an anti LC3 monoclonal antibody (3868, Cellsignal^®^, Waltham, MA, USA). Specific unconjugated proteins were detected using a 1:10,000 dilution of horseradish peroxidase-conjugated goat anti-Rabbit IgG (A16104, Thermofisher^®^, Waltham, MA, USA) and visualized by an enhanced chemiluminescence system (Advansta Diagnostics, Menlo Park, CA, USA).

A part of each tumor was treated with formalin for histological analysis. After paraffin inclusion, transversal sections of 5 μm were made. Vital coloration with hematoxylin and eosin was applied to tumoral tissue. Immuno-staining for Caspase 3 antibody (32351Abcam^®^, Cambridge, MA, USA), revealed with an anti-Rabbit antibody, was realized on 3 tumoral sections of each group. Paw skin was also included in paraffin after formalin treatment, and a histological analysis with Luxol Fast Blue protocol was made [58,59].

### 4.10. Statistics

Statistical analysis was performed using GraphPad Prism 5 and Excel v.16.48. Student’s *t*-test, ANOVA, and the Dunnett test were performed. The *p*-values were denoted as follows: * *p* < 0.05, ** *p* < 0.01, *** *p* < 0.001, and **** *p* < 0.0001. The artwork was created with GraphPad Prism 5 and Excel v.16.48.

## Data Availability

The datasets analyzed during the current study are available from the corresponding author upon reasonable request.

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
