# Peer review of "A New Manganese Superoxide Dismutase Mimetic Improves Oxaliplatin-Induced Neuropathy and Global Tolerance in Mice"

_ijms, 2022, doi:10.3390/ijms232112938_

Round 1

Reviewer 1 Report (New Reviewer)

This paper describes syntheses, characterizations and anti-tumoral activities of novel Pt(IV) conjugate prodrugs combining oxaloplatin and a Mn SOD mimic. This is a carefully done study and the findings are interesting. However, if the reaction of OxPt-1OH1-C1A, OxPt-1-1c1A, OxPt-2-1C1A with MnCl2 yields the hydrolysis products of Pt(IV) species and Mn(III)1C1A, then the question remains a mixture of Pt(IV) species and Mn(III)1C1A in appropriate ratio could have been used instead of OxPt-1OH-1C1A, OxPt-1-1C1A and OxPt-2-1C1A . Authors need to answer this question. Thus, this paper is worth publishing in International Journal of Molecular Science with minor revisions.

Author Response

We thank the reviewer for his review and interesting comments. As we mentioned in the text, the hydrolysis upon MnCl2 coordination is expected to yield to the Mn1C1A complex and a Pt(IV) species. However, we could not identify the Pt species formed and further reactivity leading to further hydrolysis compounds and/or Pt(II) derivatives could not be ruled out. This is the main reason why we decided to evaluate the conjugates themselves and not a combination of Pt(IV) intermediate + Mn1C1A. Including such associations as control conditions would have been interesting and we thank again the reviewer for this comment.

Reviewer 2 Report (New Reviewer)

The described research is carefully and thoroughly described while being both interesting and timely.  Ethical concerns are appropriately addressed.

The manuscript can be readily accepted upon minor revision as described below.

Figure 1 and its caption need to be on the same page.

Run-on sentences and improper comma usage (or lack thereof) are ubiquitous throughout the manuscript; punction needs to be carefully addressed.

Throughout the supporting material, careful attention needs to be placed on significant figure, e.g. "(323.0 mg, 3.5 mmol, 1.0 eq.)".

What solvent was used for ESI samples?

Author Response

We thank the reviewer for his comments and his careful reading. We will pay careful attention to the proper layout of the manuscript after transfer into the journal template, and make sure figures and corresponding captions are on the same pages. We have done our best to address punctuation issues in the manuscript and hope we have achieved a satisfactory improvement.

We have corrected some significant decimal digits in the supporting information.

The solvent used for ESI samples was either methanol or acetonitrile.

This manuscript is a resubmission of an earlier submission. The following is a list of the peer review reports and author responses from that submission.

Round 1

Reviewer 1 Report

The manuscript by Prieux-Klotz et al. took inspiration from the reported synergistic effect (displayed both in vitro and in clinical studies) in combination treatment with oxaliplatin and SOD-mimic Mn-complexes to design the stated conjugates. Thus, the rationale of the study is sound. The adopted biological assays are various, well-performed, and suited to the scope. Unfortunately, the reported antitumor effects do not seem fully rewarding, and do not give a definitive proof that the prodrug approach may give some advantage with respect to combination therapies. Most importantly, the authors stated the compounds were not stable (page 15, lines 358-362): ”Detailed physicochemical studies on the MAGOX series using UV-Vis titrations and 358 HPLC demonstrated an unexpected instability of these conjugates upon Mn coordination 359 of the ligand and fast degradation (t1/2 between 30 min for MAGOX-MonoOAc and 2 h for 360 MAG2OX) leading us to propose a cleavage of the Pt-O bond with the folding of the car- 361 boxylate arm onto the Mn (favoured by the formation of 5-membered metallacycle) (Fig- 362 ure 10)(30).” However, I could not find any detailed description of the synthesis, chemical characterization and stability profile of the conjugates in the manuscript, SI and cited reference, which is uncheckable (Under preparation). I believe that such detailed description should be reported in this manuscript. In any case, plasma stability is an important pharmacokinetic prerequisite for prodrug conjugates. Thus, an evaluation of the chemical stability of the conjugates on more appropriate models is needed (e.g., stability assays performed by LC-DAD-MS/MS analysis). Furthermore, both the introduction, results and discussions sections can be improved.

Minor points:

Please reduce the length of the abstract to make it more concise.

Please add the chemical structures of Mangafodipir and Calmangafodipir in Fig 1. 

Page 16, line 386 needs reference(s)

Author Response

Reviewer 1

The manuscript by Prieux-Klotz et al. took inspiration from the reported synergistic effect (displayed both in vitro and in clinical studies) in combination treatment with oxaliplatin and SOD-mimic Mn-complexes to design the stated conjugates. Thus, the rationale of the study is sound. The adopted biological assays are various, well-performed, and suited to the scope. Unfortunately, the reported antitumor effects do not seem fully rewarding, and do not give a definitive proof that the prodrug approach may give some advantage with respect to combination therapies. Most importantly, the authors stated the compounds were not stable (page 15, lines 358-362): ”Detailed physicochemical studies on the MAGOX series using UV-Vis titrations and 358 HPLC demonstrated an unexpected instability of these conjugates upon Mn coordination 359 of the ligand and fast degradation (t1/2 between 30 min for MAGOX-MonoOAc and 2 h for 360 MAG2OX) leading us to propose a cleavage of the Pt-O bond with the folding of the carboxylate arm onto the Mn (favoured by the formation of 5-membered metallacycle) (Figure 10)(30).” However, I could not find any detailed description of the synthesis, chemical characterization and stability profile of the conjugates in the manuscript, SI and cited reference, which is uncheckable (Under preparation). I believe that such detailed description should be reported in this manuscript.

Thank you for this remark. The development of these Pt(IV) conjugates, the detailed analysis of their chemical stability (combining UV-vis spectrometry, HPLC, and Mass spectrometry investigations) and the subsequent design of a stabilized series will be detailed in an independent manuscript that is still under preparation. We believe this study will be of primary interest on its own for the community of inorganic chemists and platinum chemists. Its description was not the purpose of the present work and was therefore not included. Not to blur the message of the present manuscript, we modified it by removing Figure 10.

In any case, plasma stability is an important pharmacokinetic prerequisite for prodrug conjugates. Thus, an evaluation of the chemical stability of the conjugates on more appropriate models is needed (e.g., stability assays performed by LC-DAD-MS/MS analysis.)

Thank you for this remark. We agree that plasma stability is an important parameter. We characterized the stability in buffer, in sodium ascorbate reducing medium (used to evaluate the reduction of Pt(IV) to Pt(II)) and in culture medium. Upon degradation, a liberation of the Mn1C1A complex is observed along with a platinum species leading to an oxaliplatin derivative.

Furthermore, both the introduction, results and discussions sections can be improved.

Thank you. A reorganization of the results, discussion and introduction was made.

Minor points:

Please reduce the length of the abstract to make it more concise. Thank you for your remark. The abstract word count has been decreased to 325 words.

Please add the chemical structures of Mangafodipir and Calmangafodipir in Fig 1. Thank you for this remark. We added the chemical structures of mangafodipir and calmangafodipir to the Figure 1.

Page 16, line 386 needs references. Thank you for your remark. We added references 41-43.

Reviewer 2 Report

Manuscript "A new Manganese superoxide dismutase mimetic improves oxaliplatin induced neuropathy and global tolerance on mice" describe design and study of new Mn SOD mimetic called Mn1C1A associated with oxaliplatin showed superior anti tumoral activity in vitro than oxaliplatin. The article can be accepted after the authors correct some of the shortcomings:

The authors focus only on biological studies, without mentioning in the article what methods were used to prove the structure of the obtained complexes.

The abstract is too long, it is not necessary to briefly prescribe each section in it. It is enough to briefly characterize the main results of the work

Author Response

Reviewer 2

Manuscript "A new Manganese superoxide dismutase mimetic improves oxaliplatin induced neuropathy and global tolerance on mice" describe design and study of new Mn SOD mimetic called Mn1C1A associated with oxaliplatin showed superior anti tumoral activity in vitro than oxaliplatin. The article can be accepted after the authors correct some of the shortcomings:

The authors focus only on biological studies, without mentioning in the article what methods were used to prove the structure of the obtained complexes.

Thank you for this remark. The development of these Pt(IV) conjugates, the detailed analysis of their chemical stability (combining UV-vis spectrometry, HPLC, and Mass spectrometry investigations) and the subsequent design of a stabilized series will be detailed in an independent manuscript that is still under preparation. We believe this study will be of primary interest on its own for the community of inorganic chemists and platinum chemists. Its description was not the purpose of the present work and was therefore not included.

The compounds were fully characterized using NMR (1H, 13C) and high resolution mass spectrometry, their purity was assessed by HPLC (purity >95%), and Mn coordination characterized by UV-Vis spectrometry

The abstract is too long, it is not necessary to briefly prescribe each section in it. It is enough to briefly characterize the main results of the work. Thank you for your remark. The abstract word count has been decreased to 325 words.

Round 2

Reviewer 1 Report

The authors comply with most of my suggestions. However, they did not address the main point raised in previous revision (i.e., no detailed description of the synthesis, chemical characterization and stability profile of the conjugates). In my opinion, it is not correct to show biological results about such conjugates without detailing chemical synthesis and proper characterization. I would suggest to first publish synthesis, chemical  characterization and stability profile of the conjugates, and then the biological results. However, if the Editor agree with your reply, I will rely on his/her decision. 

Author Response

Thank you for your remark. We added some bibliographic references and précisions in the Methods (already published data on MAG synthesis, stability testing). The detailed Chemistry part was not the purpose of this Study. We wanted to show the Clinical relevance of MnSOD mimetic associated with oxaliplatin to lower its toxicity.  

Round 3

Reviewer 1 Report

The synthesis, chemical characterization and stability profile of the conjugates must be provided, otherwise the manuscript cannot be accepted for publication in its current form.